# Cultured Mesenchymal Cells from Nasal Turbinate as a Cellular Model of the Neurodevelopmental Component of Schizophrenia Etiology

**DOI:** 10.3390/ijms242015339

**Published:** 2023-10-19

**Authors:** Victoria Sook Keng Tung, Fasil Mathews, Marina Boruk, Gabrielle Suppa, Robert Foronjy, Michele T. Pato, Carlos N. Pato, James A. Knowles, Oleg V. Evgrafov

**Affiliations:** 1Department of Cell Biology, State University of New York at Downstate, Brooklyn, NY 11203, USA; 2Department of Otolaryngology, State University of New York at Downstate, Brooklyn, NY 11203, USA; 3Department of Psychiatry, Rutgers University, Piscataway, NJ 08854, USAcarlos.pato@rutgers.edu (C.N.P.); 4Human Genetics Institute of New Jersey, Rutgers University, Piscataway, NJ 08854, USA; knowles.j@rutgers.edu

**Keywords:** schizophrenia, neurodevelopment, mesenchymal cells, scRNA-seq, middle turbinate

## Abstract

The study of neurodevelopmental molecular mechanisms in schizophrenia requires the development of adequate biological models such as patient-derived cells and their derivatives. We previously utilized cell lines with neural progenitor properties (CNON) derived from the superior or middle turbinates of patients with schizophrenia and control groups to study schizophrenia-specific gene expression. In this study, we analyzed single-cell RNA seq data from two CNON cell lines (one derived from an individual with schizophrenia (SCZ) and the other from a control group) and two biopsy samples from the middle turbinate (MT) (also from an individual with SCZ and a control). We compared our data with previously published data regarding the olfactory neuroepithelium and demonstrated that CNON originated from a single cell type present both in middle turbinate and the olfactory neuroepithelium and expressed in multiple markers of mesenchymal cells. To define the relatedness of CNON to the developing human brain, we also compared CNON datasets with scRNA-seq data derived from an embryonic brain and found that the expression profile of the CNON closely matched the expression profile one of the cell types in the embryonic brain. Finally, we evaluated the differences between SCZ and control samples to assess the utility and potential benefits of using CNON single-cell RNA seq to study the etiology of schizophrenia.

## 1. Introduction

Schizophrenia (SCZ) is a brain disease with a complex etiology that is commonly presented during adolescence or early adulthood. It is widely considered that alterations in brain development play a significant role in the etiology of the disease [1]. While post-mortem brain samples can be used to investigate epigenetic, transcriptomic, and proteomic alterations in SCZ patients’ brains, examining disease-specific alterations in neuronal cells during embryonic and fetal brain development requires the use of cellular models.

Patient-derived induced pluripotent stem cells (iPSCs) can be differentiated into neural progenitors and further into neurons, and these cells can serve as cellular models to study the neurodevelopmental aspects of SCZ and other disorders [2,3,4].

An alternative approach to study the molecular processes affecting neurodevelopment in patients with brain disorders is to use cells derived from the olfactory neuroepithelium (ON), where neurogenesis occurs throughout life. The brain and ON develop from neighboring ectoderm regions, and the neural crest also contributes somewhat [5,6]. Although the brain undergoes a more advanced developmental process, we may expect that, during the early stages of development, the brain will still share some of the cell types of the ON due to their common origin (the neuroepithelium). The ability of ON to produce neuronal cells suggests that the similarity between them could be substantial, at least in some cell types.

The hypothesis that ON could serve as a model to study SCZ is further supported by the strong correlation between SCZ and anosmia, affecting neuronal functions in the brain and ON, respectively, and findings regarding the dysregulation of olfactory neuron lineages in SCZ [7].

Several cellular models based on patient-derived cells from the olfactory mucosa have been developed using different protocols [8,9,10,11,12,13]. In our previous studies, we used a protocol for developing cell cultures from ON that was originally proposed by Wolozin et al. [8]. The key element of this protocol is to cover small pieces of ON with Matrigel and propagate only those cells that penetrate through the gel (see protocol details in [14]). We further demonstrated that this protocol resulted in the propagation of cells of a single cell type, the expression pattern of which remained consistent in a number of successive passages.

We named these cells CNON (Cultured Neuronal Cells derived from the Olfactory Neuroepithelium) and considered them neural progenitors, although their ability to differentiate into neurons in vivo has not been proven. We subsequently discovered that the same cell type could be generated from both superior and middle turbinate [15].

We previously developed CNON from 256 individuals, including 144 patients with schizophrenia (SCZ), and demonstrated the robustness of CNON development and the consistency of expression profiles during growth in culture and between individuals [14]. Using single-cell transcriptomics, we identified a cell type in middle turbinate (MT) with an expression profile corresponding to CNON [16], confirming that CNON is not a mixture of cell types but instead a single cell type with a specific gene expression profile. The aim of this study is to assess the similarity of CNON to cells in the middle turbinate, the olfactory epithelium, and the brain using single-cell transcriptomics and to evaluate the relevance of this cellular model in the study of brain disorders.

## 2. Results

To assess the utility of cultured cells derived from nasal turbinates to study the developmental mechanisms of schizophrenia, we performed scRNA-seq of cultures derived from a patient with SCZ (CNON-SCZ, male, Caucasian, 61 years old at the time of biopsy) and from an individual from the control group (CNON-CTRL, male, African American, 62 y.o). To identify parental cell types, we compared these data with scRNA-seq data from two middle turbinate biopsy samples taken from a patient with SCZ (MT-SCZ) and a control (MT-CTRL) (both females, African American, 58 and 32 y.o, respectively). Lastly, CNON datasets were compared with single-cell transcriptome data from an embryonic brain [17].

### 2.1. Characterization of CNON Cell Type

The Mesenchymal and Tissue Stem Cell Committee of the International Society for Cellular Therapy has developed a set of minimum criteria for defining multipotent mesenchymal stromal cells: (a) the expression of certain proteins/genes, (b) the ability to adhere to plastic, and (c) the ability to differentiate in vitro into osteoblasts, adipocytes, chondroblasts [18]. CNON satisfied the first two criteria of expressing specific genes and their ability to adhere to plastic (Figure 1B), but we have yet to investigate the multipotency of CNON.

In previous studies, WNT signaling has been shown to play an important role in controlling both the maintenance and differentiation of mesenchymal stem cells [19,20,21,22]. The CNON expression profile shows a robust expression of Wnt5A, Wnt5B, and Notch2 ligands, accompanied by genes involved in corresponding signaling pathways. For example, genes for frizzled receptors *FZD2* and *FZD7*, co-receptors *ROR2*, *LPR5*, *LPR6*, and secreted frizzled receptors *SFRP1* and *SFRP2*, as well as *CTNNB1* (β-catenin), are robustly expressed in CNON to support both canonical and non-canonical Wnt signaling (Appendix A). Similarly, *JAG1*, presenilins *PSEN1* and *PSEN2*, *ADAMS17*, *PSENEN*, and *APH1A* (γ-secretase subunits) are also expressed in CNON (Appendix A). This suggests that CNON can regulate cell differentiation and function using Wnt and Notch signaling in an autocrine or paracrine manner.

### 2.2. Comparison CNON with Middle Turbinate Dataset

We employed *reference mapping* from Seurat, which mapped query datasets (CNON) to MT reference dataset and assessed if the transcriptome of each CNON cell could be assigned to any cluster/cell type of the reference dataset. The quality of the assigned cell type of each query cell was evaluated using a calculated *prediction score*.

Such reference mapping showed that the vast majority of CNON-CTRL cells (11,741, 96%) mapped to the cluster corresponding to the Mesenchymal cell (MC) cluster of the MT-CTRL, with an average prediction score of 0.96. However, cells from the MC cluster did not proliferate (did not express cell cycle-specific genes such as *MKI67*), and the effect of cell cycle genes may reduce the quality of mapping to non-proliferating cells of the same type. To mitigate the effects of CNON proliferation, we assigned cells with a cell cycle score based on the expression of canonical cell cycle genes, applied these scores to model the relationship between gene expression and cell cycle score, and removed this cell cycle phase variability from our data using regression. Indeed, regressing cell cycle score resulted in the accurate mapping of all CNON-CTRL cells to the MC cluster, with a perfect prediction score of 1 (Table 1).

Figure 2A,B show the results of mapping CNON data from both samples onto two MT datasets after applying Uniform Manifold Approximation and Projection (UMAP) for dimension reduction [23]; all six figures are presented using the same axes and on the same scale. The cell types in MT-CTRL were annotated based on marker genes specific for every cluster. Such genes were identified via differential gene expression analysis using a previously described protocol [24]. The heatmap (Figure 2C) of MT-CTRL reveals a clear distinction between various cell types. While Figure 2A,B demonstrate the allocation of CNON to the cluster MC in UMAP coordinates, Table 1 provides the number of cells mapped to specific clusters with corresponding prediction scores, supporting our previous findings that CNON was developed from one cell type. A comparison of marker genes across different DEX analyses further confirmed that the cell cultures originated from only one cell type of MT (Figure 3A).

Differential expression (DEX) analysis revealed that transcripts of MC markers are found in the vast majority of CNON cells. Meanwhile, expressions of prominent markers of other middle turbinate cell types, including basal (*SERPINB3*, *KRT5*), endothelial (*CCL14*, *VWF*), serous (*DMBT1*), club (*LYPD2*, *SCGB1A1*), ciliated (*SNTN*), goblet cells (*MUC5B*), and ionocytes (*CFTR*), were either insignificant or not present, as they were identified in less than 1% of CNON (Table 2).

### 2.3. Characterization of MC Cell Type

To better characterize the MC cell type, we performed a gene ontology enrichment analysis on the differentially expressed (DEX) genes in MC cells (adjusted *p*-value < 0.05) and compared the results to all other cell types in the middle turbinate. The analysis revealed significant enrichment in multiple biological processes related to development (Appendix A). Other noteworthy biological processes include those related to cell adhesion, cell migration, and mesenchymal cell maintenance (mesenchymal development, mesenchymal cell differentiation, regulation of epithelial to mesenchymal transition, positive regulation of epithelial to mesenchymal transition, epithelial to mesenchymal transition). While there was an enrichment of genes involved in neurogenesis (GO:0022008) and related processes (generation of neurons (GO:0048699), regulation of neuron projection development (GO:0010975), neuron projection development (GO:0031175), and neuron projection morphogenesis (GO:0048812), etc.), processes involved in the development of some non-neuronal tissues and organs were also present.

### 2.4. Comparison of Single-Cell CNON Data with the Olfactory Neuroepithelium Dataset

We mapped CNON data to single-cell datasets from the olfactory neuroepithelium [25]. This dataset contains data from four patients. We focused our investigation on reference mapping CNON cells to Patient 2, as their map had the most neuronal cells per sample, indicating that it accurately represents the olfactory neuroepithelium. The mapping showed that CNON-CTRL exclusively mapped to a single cluster in the data of Patient 2 (Figure 3B), and this cluster expressed markers similar to the MC cluster in MT-CTRL and MT-SCZ. The majority of cells in CNON-SCZ were also mapped to the same cluster. Additionally, we compared CNON cells with integrated data from all four patients, and almost all CNON cells mapped to a single-cell cluster (Figure 3C,D).

### 2.5. Comparison of Single-Cell CNON Data with Embryonic Brain Dataset

We then compared CNON expression profiles with a large dataset of brains at several embryonic stages of development (Carnegie stages 13–22) [17]. A dataset from the early stages of development was chosen for our comparison as we believe that the embryonic brain was more likely to contain CNON-like cells than the fetal brain at later stages of development due to a closer relationship to a common or similar ancestors.

Comparing CNON single-cell data in this study with the embryonic brain at different stages of development revealed similarity with cluster 47, as described by the authors of [17] (in the Supplementary Data). Most cells from cluster 47 were sourced from one sample, CS14_3, and we performed reference mapping of CNON data to this sample alone. Overall, 99.99% of CNON-CTRL (all except one cell) mapped solely to cluster 9 of CS14_3 (Figure 2D, Table 1), with an average prediction score of 0.995. This cluster corresponds to cluster 47 in the analysis of data from all embryonic samples [19]. Similarly, the reference mapping of a smaller dataset of CNON-SCZ resulted in 3297 cells (98.5%) being mapped to cluster 9, with an average prediction score of 0.985, while 6 cells were assigned to cluster 0, with a prediction score of only 0.015 (Table 1).

Our findings indicate that CNON exhibits a highly comparable expression profile with one cell type found in the developing brain. In the article describing the embryonic brain dataset [17], cluster 47 was classified as “others”, and it was distinctly different from explicitly classified cell types such as neurons, radial glia, neuroepithelial, intermediate progenitors, and mesenchymal cells. In sample CS14_3, this cell type accounted for about 3.5% of the total population of cells, while in other samples, including those from earlier samples (CS13), another sample from CS14, and samples from later stages (CS15, CS20, and CS22), their proportion was lower. According to our calculations, these cells make up 0.87% of all cells, while the authors estimate their fraction to be even higher at 1.1%.

### 2.6. Schizophrenia vs. Control CNON Comparison

Finally, we compared the single-cell gene expression data between CNON cells (CNON-CTRL and CNON-SCZ) to assess the potential usability of scRNA-seq to study schizophrenia. The CNON-SCZ sample was selected for this study because cells from CNON-SCZ had a low growth rate and an extreme cell cycle eigengene value in our previous study [24]. It was located on the periphery of the PCA1/PCA2 map based on all or only DEX genes, suggesting that this sample may reveal schizophrenia-specific differences in expression profiles in single-cell data despite the high genetic and transcriptomic heterogeneity of both CNON samples. Most of the DEX genes identified in our previous study had low expression, with only 21 of them having transcripts per million transcripts (TPM) more than 10 and only 5 genes with TPM > 100. In single cells, we found 15 DEX genes with detected expression in more than 50% of cells. Eleven of them showed a difference in expression between the control and SCZ samples of more than 50%, with all of them being in the same direction (assumed by the DEX analysis in the bulk RNA-seq study). We initially assumed that these alterations in gene expression could explain the dramatic reduction in the proliferation rate, but single-cell data revealed a different picture. The percentage of cells in the G2 m and S cycle stages in the slow-growing CNON-SCZ sample was higher than in the fast-growing CNON-CTRL. Instead, we found that a substantially larger fraction of cells from the SCZ sample was associated with a cluster of cells with an elevated level of mitochondrial gene expression, often characteristic of apoptotic processes (Table 3). Given that apoptosis is a relatively rapid process [26], the slow growth of the SCZ sample may potentially be explained by a higher apoptosis rate.

## 3. Discussion

Cells in CNON lack prominent markers of pluripotency, such as *POU5F1* and *NANOG*, which are the hallmarks of embryonic stem cells and iPSCs. *PROM1* (CD133), a marker used for the purification of neural stem cells [27], is also not expressed in CNON. Currently, the literature on neural progenitor cells in the brain is relatively scant. One of the better-known types is radial glia (a cell that differentiates into outer radial glia and ventricular radial glia at later stages of brain development), which can directly differentiate into neurons or produce intermediate progenitor cells (IPCs). However, the expression profile of CNON does not fit into the radial glial gene expression pattern; in particular, CNON does not express the radial glial markers *SOX2*, *HES5*, *PAX6*, or *GFAP*. CNON also does not express *EOMES*, a marker of IPCs, which play an important role in producing neurons after gestation week 8 (~PFA 6 weeks).

Neuroepithelium, such as ON and the epithelium of the middle turbinate, originates mostly from the ectoderm, while cells from the MC cluster express multiple markers of mesenchymal cells or mesenchymal stem cells, which are multipotent cells of mesodermal origin (Appendix A). The precursors of these mesenchymal cells likely originate from neural crest, from which they migrate to different locations of the fetus to establish mesenchymal cell populations in bone marrow [28], adipose tissue [29], oral mucosa lamina propria [30], and several other locations (see [31] for a review), including the brain [32]. Expectedly, cluster 9 of the embryonic brain sample CS14_3 also expressed multiple mesenchymal markers.

Despite the characteristic similarities in the expression of specific marker genes, mesenchymal cells from different tissues may differ in the expression of some genes and their ability to differentiate into particular cell types in a specific environment. For example, mesenchymal-like stem cells residing in the olfactory mucosa can demonstrate the promyelination effect on oligodendrocyte precursor cells, while similar mesenchymal cells derived from bone marrow do not enhance myelination [33]. Distinctions between the MSCs from different tissues define their specific use in regenerative medicine [34,35]. Despite their similarity with embryonic brain cells, cultured mesenchymal cells from MT or ON may have some distinct properties. While cells grow in culture for several passages prior to gene expression profiling, it is possible that nasal turbinate hypertrophy or inflammation has a long-lasting effect on expression profiles, potentially confounding the results.

Other studies have also reported cells with mesenchymal properties in the ON (ecto-mesenchymal stem cells) from the lamina propria of ON [34], and some researchers have been able to culture them in vitro, albeit using different methods of developing cell cultures [36], including the generation of neurospheres. As we demonstrated in the Results section, the single-cell data from the olfactory neuroepithelium [25] showed a large group of cells with expression profiles corresponding to CNON, which express multiple mesenchymal markers. A recent single-cell study of cell cultures derived from the olfactory mucosa also showed a large group of cells referred to as “fibroblast/stromal”. However, at the time of writing, we are unable to gain access to the necessary data to assess the expression of mesenchymal markers. Notably, the aforementioned study did not utilize cells that can penetrate Matrigel; thus, the resulting culture consists of multiple cell types, which were described as fibroblast/stromal, GBC, and myofibroblasts [37].

Another study suggested that mesenchymal cells from the olfactory mucosa possess multipotency and that they can form neurospheres different from those produced by horizontal epithelial global cells [38]. Therefore, the similarity in terms of mesenchymal properties between these cells and CNON suggests that CNON is likely a derivative of the ecto-mesenchymal stem cells of the ON. CNON is a single cell type due to the specific cell culture method, while other methods that do not feature the use of Matrigel for cell type selection result in at least three different cell types growing in culture [37].

It should be noted that CNON have a more pronounced gene expression profile of mesenchymal cell markers compared to MC clusters of MT or cluster 9 from the embryonic brain sample. For example, cells from MC in MTs express HLA-DR genes, but their expression was halted when culturing them in vitro under our conditions. This is not unexpected, as the definition of mesenchymal stem cells is based on assays of cells in culture, and for a long time, there was a hypothesis that the MSC is an artifact of culturing cells in vitro (as discussed, for example, in [39]). We attribute this to the plasticity of mesenchymal cells and the changes in phenotypes and gene expression that occur according to the environment.

Indeed, the cells in the MC clusters in turbinates do not divide; in Matrigel, they change phenotype, divide, and migrate (Figure 1A) before changing their phenotype again to a classical mesenchymal phenotype when grown in 2D (Figure 1B). However, when placed inside Matrigel after multiple passages in 2D, the cells reverted to a phenotype with multiple elongated branches and organized into complex interconnected cell structures resembling a neuronal network (Figure 1C).

The role of cells with mesenchymal gene expression signatures in the early stages of brain development is not clear. Many have noticed similarities regarding MSCs with pericytes and suggested the involvement of MSCs in forming a blood–brain barrier and vascular systems. Many also appreciate their paracrine function in a niche neurovascular context, as they are involved in orchestrating the complex development of brain structures [32]. The ability of MSCs to differentiate into neurons in vitro and be engrafted in the brain also suggests the involvement of these cells in neurogenesis. In some studies, the transplantation of MSCs in brain lesion models has shown therapeutic effects [40]. The effect may be caused by the differentiation of MSCs into neurons. Alternatively, MSCs may stimulate lesion repair via a paracrine effect, i.e., secreting factors triggering reparation performed by host cells. Anyway, the important role of MSCs in the brain is well recognized, and the hypothesis that alterations in gene expression in MSCs cause SCZ is well justified.

The role of mesenchymal cells in turbinates is not known. Although they have a capability to differentiate into neurons, it is not clear if they realize this potential in the olfactory mucosa either during development, injury, or the regular replenishment of olfactory neurons. The striking similarity of CNON to cells in the embryonic brain is probably a reflection of the similarities of MSCs among tissues. MT belongs to the respiratory system, where the role of mesenchymal cells is quite pronounced. Mesenchymal cells are involved in lung development and responsible for homeostasis and tissue repair in the lungs [41]. If alterations in the gene expression of mesenchymal cells contribute to the etiology of schizophrenia, we should expect that the same changes in MSC properties can affect other organs with a substantial amount of MSCs. Indeed, a comorbidity between SCZ and lung diseases has been reported, and in both directions, schizophrenia was found to be associated with impaired lung function [42], and patients with COPD have a 10 times higher risk of psychiatric comorbidities [43]. There are reports that olfactory deficits known to be prevalent in SCZ are also found in most COPD patients [44]. We hypothesize that alterations in the properties of mesenchymal cells may contribute to a range of “mesenchymal” disorders, which include some subtypes of schizophrenia and lung diseases.

Our findings regarding the analysis of CNON indicate that WNT signaling and the regulation of WNT production (and specifically WNT5A) are involved in the etiology of SCZ [24] and fit this hypothesis well. These pathways are crucial for the self-renewal and differentiation of MSCs [20], and they also play a critical role in lung development [45], tissue regeneration [46], and the etiology of COPD [47]. Thus, alterations in these pathways could be one of common mechanisms of “mesenchymal” disorders.

Future directions. Specific molecular pathways affected by SCZ (or other brain disorders) in mesenchymal stem cells could be assessed by studying the relationship between the expression of genes using correlation analysis (WGCNA and alike) and gene expression profiles after altering the expression of certain genes of interest (using CRISPR, siRNA, etc.). The genes in these pathways could be related to GWAS data; thus, assessing their genetic contribution to mechanisms of schizophrenia related to alterations in the expression profiles of mesenchymal stem cells could be useful. Studying cell functions and properties such as proliferation, migration, adhesion, apoptosis, etc., in patient-derived and control cells can help to better relate expression data with the specifics of brain structures revealed via brain imaging.

## 4. Materials and Methods

A diagram of the study design is presented in Figure 4.

### 4.1. Biopsy Collection and Sample Preparation

Biopsies were obtained from patients without any history of sinonasal disease or surgery or immunocompromise. Tissue samples were obtained from the superior-medial region of the head of the middle turbinate; laterality was determined by ease of access. The mucosa was anesthetized and decongested with topical 1% lidocaine and 0.05% oxymetazoline. After 5 min, 0.3 mL of 1% lidocaine with 1:100,000 epinephrine was injected into the targeted mucosal site under visualization. Also, 2 mm cupped forceps were used to obtain biopsy specimens. These samples were immediately transported to the lab in Leibovitz’s L-15 Medium (Thermo Fisher Scientific, Waltham, MA, USA), supplemented with Antibiotic-Antimycotic solution (Thermo Fisher Scientific, Waltham, MA, USA), and processed to prepare the single-cell suspension. The biopsy pieces were minced with two scalpels in a Petri dish with a small amount of cold Hank’s buffer without magnesium or calcium and then transferred to a 1.5 mL Eppendorf tube and washed twice with 1 mL cold Hank’s buffer. The cells were dissociated in 250 µL of 0.25% Trypsin-0.02% EDTA (VWR, Rochester, NY, USA) at 37 °C while shaking at 500 rpm. After 10 min, the suspension was mixed by pipetting and returned to the thermomixer for another 10 min. After being subjected to mixing once more, the cells were sedimented via centrifugation at 300 relative centrifugal force for 5 min, resuspended in 500 µL of cold Hank’s solution (Thermo Fisher Scientific/Waltham, MA) with BSA, filtered using 40 µm FLOWMI cell strainer (Bel-Art, Wayne, NJ, USA) cell suspension into a 1.5 mL Eppendorf tube, centrifuged again, and finally resuspended in 50 µL of Hank’s with 0.04% BSA (Thermo Fisher Scientific, Waltham, MA, USA). 

### 4.2. CNON Cell Culture

The protocols used for developing CNON cell cultures from middle or superior turbinate biopsies have been previously described [14]. In brief, each biopsy sample was dissected into 3–4 pieces approximately 1 mm^3^ in size, and each piece was placed onto the surface of a 60 mm tissue culture dish coated with 25 µL of Matrigel Basement Membrane (Corning, Tewksbury, MA, USA) reconstituted in F12 Coon’s medium (Sigma-Aldrich/St. Louis, MO) and then covered by 15 µL of full-strength Matrigel. After the Matrigel gelatinized, 5 mL of medium 4506 [8] was added. Medium 4506 is based on F12 Coon’s medium (Sigma-Aldrich/St. Louis, MO) supplemented with 6% of FBS KSE Scientific, Durham, NC, USA), 5 ug/mL human Gibco transferrin (Thermo Fisher Scientific/Waltham, MA), 1 ug/mL human insulin (Sigma-Aldrich/St. Louis, MO), 10 nM hydrocortisone (Sigma-Aldrich, St. Louis, MO, USA), 2.5 ng/mL sodium selenite (Sigma-Aldrich, St. Louis, MO, USA), 40 pg/mL thyroxine (Sigma-Aldrich, St. Louis, MO, USA), 1% Gibco Antibiotic-Antimycotic (Thermo Fisher Scientific, Waltham, MA, USA), 150 µg /mL Bovine hypothalamus extract (MilliporeSigma, Rockville, MD, USA), and 50 µg/mL Bovine pituitary extract (Sigma-Aldrich, St. Louis, MO, USA). Within 1–4 weeks of incubation, the CNON cells were observed to grow out of the embedded pieces of tissue. Due to their unique ability to grow through Matrigel, CNON often populate large areas that are free from the presence of other cell types. Outgrown cells with a mesenchymal phenotype were then physically isolated using cloning cylinders from areas where no cells with other phenotypes were present, dislodged using 0.25% Trypsin-0.02% EDTA (VWR, Rochester, NY, USA), and transferred into a new Petri dish for further cultivation.

### 4.3. Single-Cell Preparation from CNON

At ~80% monolayer on a 6 cm Petri dish, the cells were dislodged with 1 mL of 0.25% Trypsin-0.02% EDTA solution, and 3 mL 4506 culture medium was added to stop digestion. The cells were then gently and thoroughly mixed to break up clumps of cells, spun at 300 rcf for 5 min, and resuspended in a 3 mL culture medium. After gentle mixing by pipetting, the cell suspension was filtered using a 40 µm FLOWMI cell strainer into a 15 mL tube. The cells were washed with 1× DPBS (Thermo Fisher Scientific, Waltham, MA, USA) with 0.04% BSA, then 1× PBS (Thermo Fisher Scientific, Waltham, MA, USA) with 0.04% BSA, re-suspended in 500 μL of PBS, and filtered using Flowmi™ Tip Strainer. After counting, the cell concentration was adjusted to 700 cells/μL.

### 4.4. scRNA-seq Sample Processing

The concentration and viability of the cells were gauged using a hemocytometer and Trypan blue. After counting, single-cell libraries were prepared according to the 10× Genomics protocol CG000183 on Chromium controller (10× Genomics) and sequenced on NovaSeq6000 as paired-end 28 + 90 bp reads plus two indexing reads.

Raw sequencing data were processed using bcl2fastq2 v2.20 to convert the BCL files to fastq files while simultaneously demultiplexing. The fastq files were processed using TrimGalore v. 0.6.5 to automate quality and adapter trimming and perform quality control.

### 4.5. scRNA-seq Data Analysis

Subsequent fastq processing was carried out using Cell Ranger v.6.1.2 (10× Genomics) to generate raw gene–barcode matrices from the reads, which were aligned to the GRCh38 Ensembl v93-annotated genome. Cell Ranger utilized the processed FASTQ files to perform alignment, filtering, barcode counting, and UMI counting. Then, the filtered feature–barcode matrix, a Cell Ranger output containing only detected cell-associated barcodes, was used as an input for the Seurat R package (V4.1.0) [48].

Using R package Seurat v.4.10, cells with a number of detected RNA molecules or genes less than 3 were filtered out from the data. Then, the gene expression was normalized and scaled using Sctranform(V1), a variance-stabilizing transformation method that employs a regularized negative binomial regression model and provides rigorously takes into account technical biases.

To account for any potential confounding effect due to cell cycle phase, we calculated cell cycle phase scores using Seurat’s built-in lists of cell cycle genes. We then regressed these scores from the dataset during normalization in order to reduce the effect of cell cycle heterogeneity on our analysis using the Seurat algorithm (https://satijalab.org/seurat/articles/cell_cycle_vignette.html, accessed on 10 March 2022).

Then, we identified clusters of cells based on their gene expression profiles using a graph-based clustering algorithm. Before clustering, we reduced the dimensionality of the data using Principal Component Analysis (PCA), and the first 30 principal components were retained. Using the PCA results, the nearest neighbors for each cell were determined, which were subsequently used in graph-based clustering to segregate cells into different clusters. After cluster tree analysis (clustree), we selected an optimal resolution parameter (0.5 for MT studies) for cluster analysis. We annotated the clusters using known markers and data from relevant single-cell studies and generated UMAP (Uniform Manifold Approximation and Projection) plots for visualization.

To identify genes that are differentially expressed (DEX) between the cell clusters in scRNA-seq data, we used the FindAllMarkers() function in Seurat. This function compares the expression of each gene in each cluster using a Wilcoxon rank-sum test and returns a list of differentially expressed genes. This non-parametric test was chosen for its minimal assumptions about the underlying data distribution. To account for the multiple comparisons problem in transcriptome-wide analyses, *p*-values were adjusted using Bonferroni correction. We set the min.pct argument to 0.25 (to include only genes detected in at least 25% of cells in one of groups in comparison) and the logfc.threshold argument to 0.25 (log fold change of at least 0.25). To visualize the differential expression (DEX) results, we used the DotPlot() function in Seurat, which plots the fold change in gene expression using the color scale and the size of the circle as percentage of cells expressing the gene.

To compare the expression profiles of CNON cells with the single-cell data from each tissue (the middle turbinate, olfactory neuroepithelium, and embryonic brain), we performed reference mapping using Seurat. This allowed for the transfer of cellular annotations from the middle turbinate, olfactory neuroepithelium, and embryonic brain datasets (reference datasets) to the CNON datasets (query datasets). We calculated and identified the shared canonical correlation between each reference dataset and each of the CNON datasets in each round of reference mapping using FindTransferAnchors(). The function’s parameters were optimized to utilize Sctranform and top 30 principal components. We also set the k-nearest neighbor approach to 5, which allowed us to identify similar cells based on their expression profiles. Upon establishing the anchors, cell type annotations from the reference dataset were mapped onto each CNON dataset. This was carried out using TransferData() to evaluate the similarity of each cell found in the CNON datasets to their most similar counterparts in the reference datasets in terms of gene expression profile. To show the accuracy and reliability of the transferred annotations in the CNON datasets from each reference datasets, we calculated the prediction score for each cell type annotation in Seurat, which is a normalized measure that captures the cumulative weights (from the anchors) that a query cell belongs to a specific cluster in the reference dataset. ScRNA-seq data from the embryonic brain were obtained from the UCSC Cell Browser (matrixes from combined samples) and the NeMO repository (https://assets.nemoarchive.org/dat-0rsydy7, accessed on 20 January 2022), where individual sample matrixes were available. The ScRNA-seq data from olfactory neuroepithelium were obtained from Gene Expression Omnibus under accession code **GSE139522**.

## 5. Conclusions

Through using scRNA-seq, we confirmed that CNON cells originate from a single cell type of the middle turbinate or the olfactory neuroepithelium. The expression profile of CNON closely matches that of the mesenchymal stem cells. Although we have not yet tested the multipotency of these cells, other studies suggest that mesenchymal cells of the olfactory mucosa are able to differentiate into multiple lineages, including neurons.

Our analysis of gene expression in the embryonic brain [17] helped to identify a cell type that closely matches CNON cells in terms of expression profiling. The cell type homogeneity of CNON, stability of their expression profiles in cell culture during multiple passages, and high similarity to one of the cell types in the embryonic brain support the notion that CNON can be used to study alterations in gene expression and the functions of mesenchymal stem cells in brain development and provide a cellular model of neurodevelopmental disorders.

## Figures and Tables

**Figure 1 ijms-24-15339-f001:**
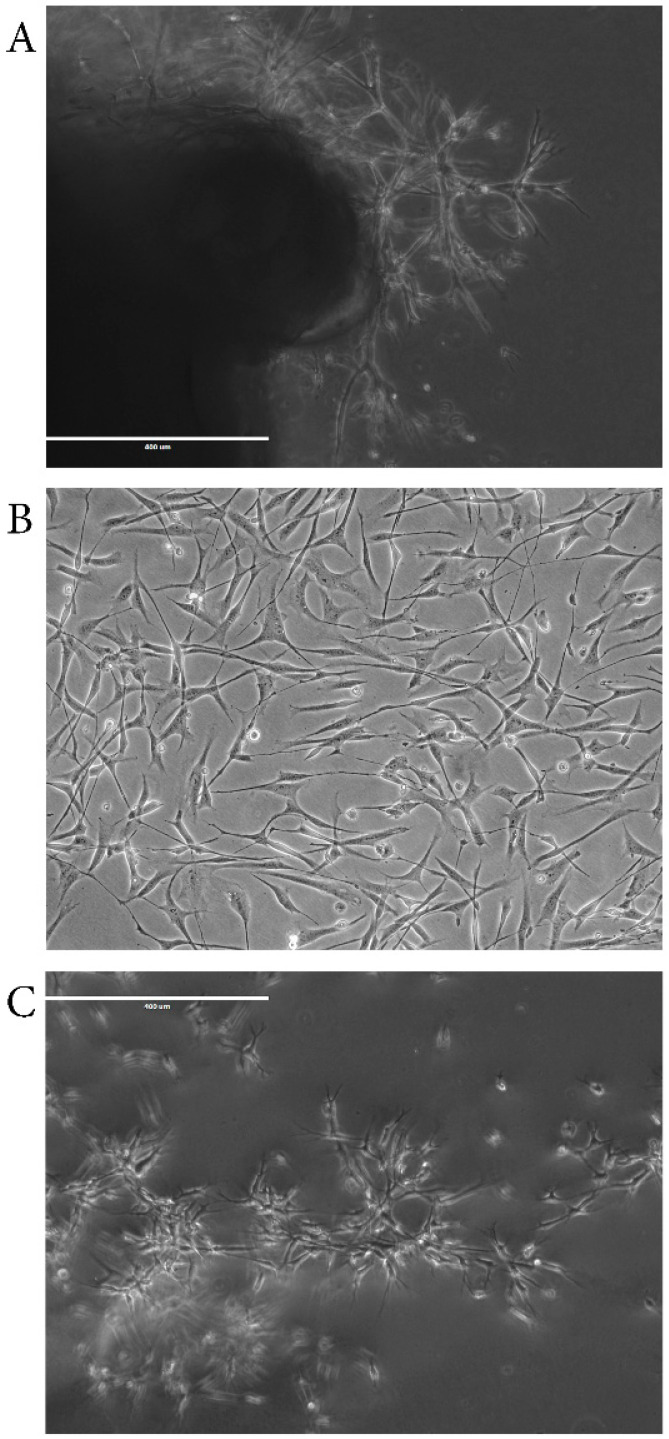
Microscopic images of CNON cell (**A**) outgrowing from biopsy piece in Matrigel, (**B**) growing in 2D culture, and (**C**) growing in Matrigel after 2D culturing for several passages. All pictures are presented at the same magnification (objective 10×).

**Figure 2 ijms-24-15339-f002:**
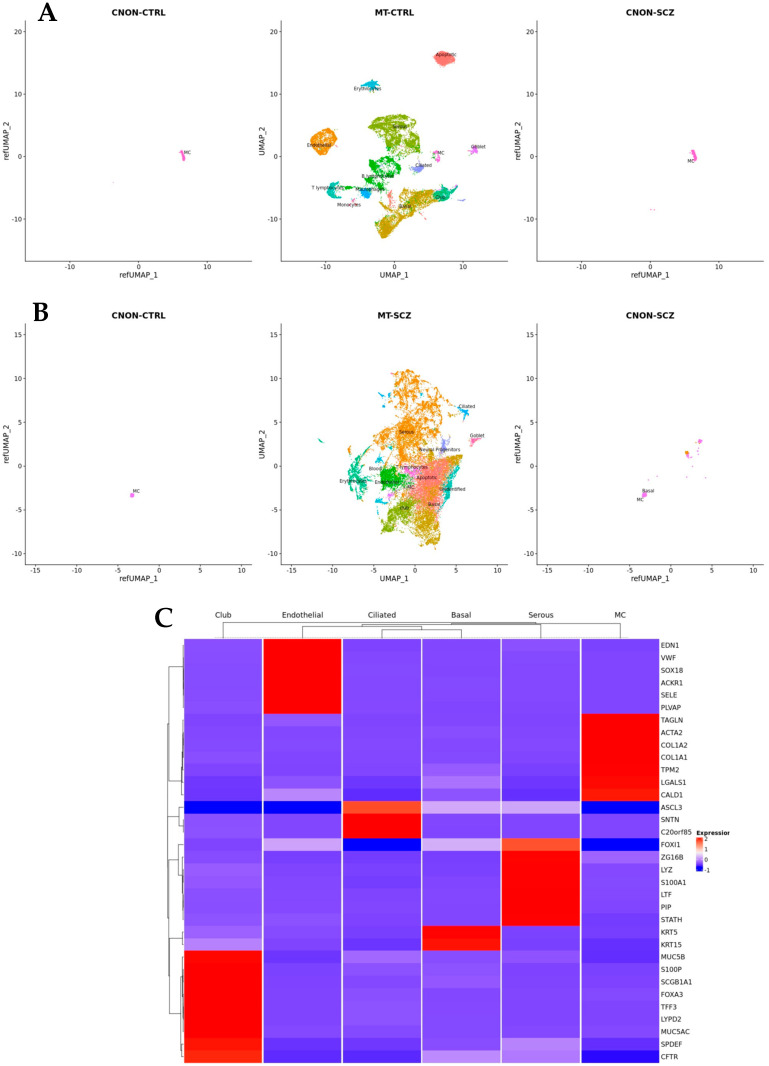
Single-cell reference mapping of CNON datasets (CNON-CTRL and CNON-SCZ) to human middle turbinate datasets (MT-CTRL and MT-SCZ) with cell cycle regression. (**A**) UMAP dimensionality reduction plot of 21,565 MT-CTRL cells displaying 13 distinct cell types (**central panel**). All cells from CTRL (**left panel**) and SCZ (**right panel**) are mapped to the MC cluster in MT-CTRL. All three datasets are shown in the same UMAP coordinates as MT-CTRL. (**B**) UMAP dimensionality reduction plot of 28,140 MT-CTRL cells with the same 13 annotated cell types as in MT-CTRL (**central panel**). CNON-CTRL cells are mapped exclusively to MC cluster (**left panel**); 3268 CNON-SCZ cells are mapped to MC cluster, while 35 cells are mapped to Basal cluster. (**C**) Heatmap representation of marker genes across different MT cell types.

**Figure 3 ijms-24-15339-f003:**
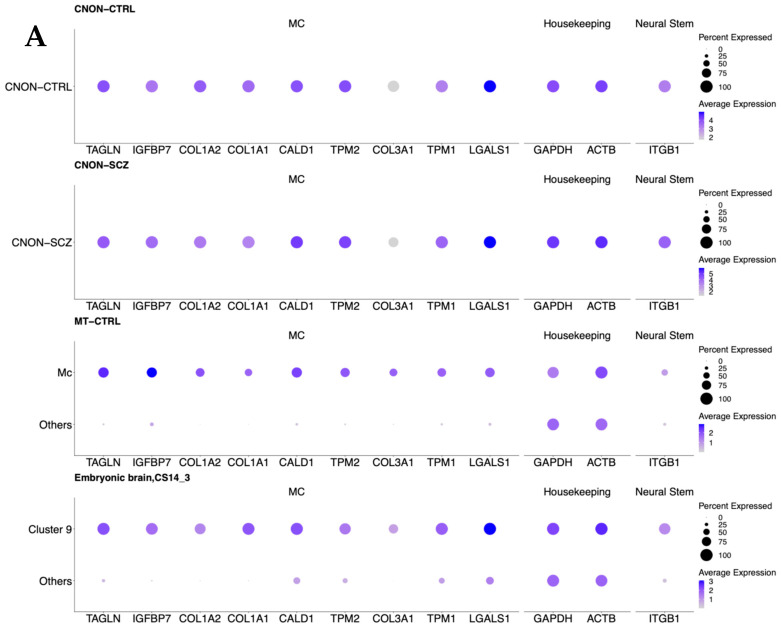
(**A**) Average gene expression of selected cell markers in CNON. A gene is considered a MC marker gene if it is expressed in the MC cluster higher than in all other clusters with (a) statistical significance, (b) logFoldChange > 2, and (c) expressed in at least 50% of cells of MC. MC markers *TAGLN*, *COL1A2*, *COL1A1*, *CALD1*, *TPM2*, *COL3A1*, *TPM1*, and *LGALS1*; housekeeping markers *GAPDH* and *ACTB*; and the neural stem cell marker *ITGB1* are shown in CNON-CTRL, CNON-SCZ, MT-CTRL, and the embryonic brain (sample CS14_3). The size of the dot represents the percentage of cells expressing the gene, and the colors represent the average expression level of each gene. (**B**) Single-cell reference mapping of CNON datasets to the olfactory neuroepithelium, Patient 2. **Central panel** shows UMAP dimensionality reduction plot of cells from the olfactory neuroepithelium of Patient 2 (25) with 13 cell types annotated. **Left panel**: UMAP dimensionality reduction plot of 11,425 CNON cells (CNON-CTRL); all cells mapped to the mesenchymal cell type. **Right panel**: UMAP dimensionality reduction plot of 2547 CNON cells (CNON-SCZ). The majority of cells (2420 CNON-SCZ cells) are mapped to the mesenchymal cell type, and 127 CNON-SCZ cells are mapped to the vascular smooth muscle cell clusters. (**C**) Single-cell reference mapping of CNON datasets to olfactory neuroepithelium dataset (integration of four patient sample data). **Central panel** shows UMAP dimensionality reduction plot of cells from the olfactory neuroepithelium, integrated data from four patients (25) with 13 cell types annotated. **Left panel**: UMAP dimensionality reduction plot of 10,979 CNON cells (CNON-CTRL); 10901 cells are mapped to the mesenchymal cell type, while 78 cells are mapped to the vascular smooth muscle cell clusters. **Right panel**: UMAP dimensionality reduction plot of 2075 CNON cells (CNON-SCZ). The majority of cells (1868 CNON-SCZ cells) are mapped to the mesenchymal cell type, and 207 CNON-SCZ cells are mapped to the vascular smooth muscle cell clusters. (**D**) Single-cell reference mapping of CNON datasets to embryonic brain (CS14_3). **Central panel**: UMAP dimensionality reduction plot of CS14_3 with 13 clusters. **Left panel**: UMAP dimensionality reduction plot of 12,234 CNON cells (CNON-CTRL); 12,233 cells are mapped to Cluster 9, 1 cell is mapped to cluster 0. **Right panel**: UMAP dimensionality reduction plot of 3303 CNON cells (CNON-SCZ). 3297cells are mapped to cluster 9, and 6 cells are mapped to cluster 0.

**Figure 4 ijms-24-15339-f004:**
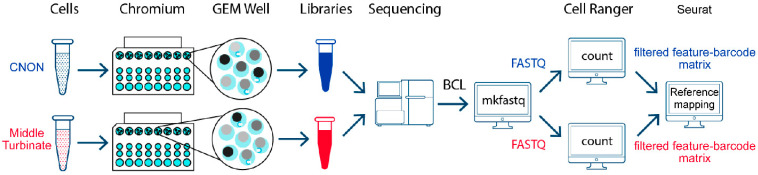
Diagram of study design.

**Table 1 ijms-24-15339-t001:** Mapping CNON cells onto various reference datasets. Number of cells from CNON-CTRL and CNON-SCZ mapped onto different cell types or clusters found in MT-CTRL (after cell cycle score regression), MT-SCZ (after cell cycle score regression), CS14_3, ON-Patient 2, and ON-Integrated, and average predicted identity scores for each cell cluster mapping.

		Predicted Number of Mapped Cells		Average Predicted ID Score	
		CNON-CTRL	CNON-SCZ	CNON-CTRL	CNON-SCZ
**MT-CTRL Cell-Cycle Regressed**	**MC**	12,234	3303	1	1
**MT-SCZ Cell-Cycle Regressed**	**Basal**	0	35	0	0.035
	**MC**	12,234	3268	1	0.964
**ON-Patient 2**	**MC**	11,425	2428	1	0.0853
	**Vascular Smooth Muscle**	0	127	0	0.117
**ON-Integrated**	**MC**	10,901	1868	0.986	0.864
	**Vascular Smooth Muscle**	78	207	0.014	0.136
**CS14_3**	**Cluster 0**	1	6	0.05	0.015
	**Cluster 9**	12,233	3297	0.995	0.985

**Table 2 ijms-24-15339-t002:** Expression of marker genes of major respiratory epithelial cell types in CNON (CNON-CTRL and CNON-SCZ), percentage of CNON cells and bulk CNON (average transcripts per million transcripts from 255 CNON samples) expressing these markers.

		CNON-CTRL		CNON-SCZ		Bulk CNON
		Average Expression	Percentage of Cells Expressing Gene	Average Expression	Percentage of Cells Expressing Gene	TPM
**Housekeeping**	**ACTB**	2.456	100	2.456	100	1540.14
	**GAPDH**	1.697	100	3.4082	100	1153.02
**Basal**	**SERPINB3**	0.00375	0.0172	0	0	0.01
	**KRT5**	0	0	0.0005	0.303	0.21
**Endothelial**	**CCL14**	0	0	0	0	0.20
	**VWF**	0.0009	0.058	0.0031	0.182	0.09
**Serous**	**DMBT1**	0.0001	0.0007	0.006	0.394	0.04
**Club**	**LYPD2**	0	0	0	0	0.02
	**SCGB1A1**	0	0	0	0.003	0.00
**Ciliated**	**SNTN**	0.0005	0.0327	0.004	0.272	0.23
**Goblet**	**MUC5B**	0	0	0.001	0.0606	0.06
**Ionocytes**	**CFTR**	0	0	0.0005	0.0303	0.07

**Table 3 ijms-24-15339-t003:** Number and percentage of cells in G0, G1, S, G2/M, and apoptotic clusters in CNON-CTRL and CNON-SCZ cell cultures.

	CNON-CTRL		CNON-SCZ	
	Number of Cells	Percentage of Cells	Number of Cells	Percentage of Cells
G0	10854	88.7	2553	77.3
G1	634	5.2	348	10.5
S	556	4.5	242	7.3
G2-M	143	1.2	106	3.2
Apoptotic	47	0.4	54	1.6

## Data Availability

The data discussed in this publication have been deposited in NCBI’s Gene Expression Omnibus [49] and are accessible through GEO Series accession number GSE219165 (https://www.ncbi.nlm.nih.gov/geo/query/acc.cgi?acc=GSE219165, accessed on 5 December 2022). Original R scripts are available from the corresponding author, VSKT, upon reasonable request.

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
