# Peer review of "Cultured Mesenchymal Cells from Nasal Turbinate as a Cellular Model of the Neurodevelopmental Component of Schizophrenia Etiology"

_ijms, 2023, doi:10.3390/ijms242015339_

Round 1

Reviewer 1 Report

Introduction:

  • The introduction provides good background but could be made more concise by reducing some repetitive details.

  • The research aims and hypotheses could be stated more explicitly in 1-2 sentences.

  • line 49, nasal turbinated could be innterested by different hpyertrophic and inflammatory disorders. Please discuss as potential limitations. discuss and cite doi:10.23812/19-522-L-4

Methods:

  • Including a diagram showing an overview of the study design, sample processing workflow, data generation and analysis would help give a clear picture.

  • More details are needed on bioinformatic and statistical analysis approaches used for the scRNA-seq data.

  • Information on sample size and power calculations is lacking and should be included.

  • Describe any measures taken to reduce technical biases in sample processing and analysis.

Results:

  • The order of presenting results could be improved for clarity - consider grouping by sample type.

  • Use figures/tables to summarize key results instead of just describing in text.

  • Report exact p-values and confidence intervals for group comparisons.

Discussion:

  • Relate current findings to previous studies identifying transcriptional profiles of nasal epithelium or embryonic brain cells.

  • line 356, snp polymorphisms associated with different phenotipic xpression should be discussed . discuss and cite doi:10.1136/bjophthalmol-2021-319756. and doi:10.1111/coa.13870.
  • Elaborate on limitations related to sample heterogeneity, sample size, technical factors etc.

  • Suggest future research directions to build on these findings.

Conclusions:

  • Restate purpose and key findings briefly to lead into conclusions.

  • Emphasize implications of the findings for modeling neurodevelopmental processes.

  • Suggest applications of the CNON model based on the results.

no

Author Response

Introduction:

The introduction provides good background but could be made more concise by reducing some repetitive details.

We remove several unnecessary phrases making the introduction more concise.

The research aims and hypotheses could be stated more explicitly in 1-2 sentences.

We stated the aim of the study in the last sentence.

line 49, nasal turbinated could be innterested by different hpyertrophic and inflammatory disorders. Please discuss as potential limitations. discuss and cite doi:10.23812/19-522-L-4

We added this point to discussion. However, we believe that the recommended citation of the study “Long-term results of nasal surgery: comparison of mini-invasive turbinoplasty” is not related to this point or to the topic of the manuscript and may mislead the readers.

Methods:

Including a diagram showing an overview of the study design, sample processing workflow, data generation and analysis would help give a clear picture.

We added a diagram of the study design. This is the first time we made such a diagram for the manuscript and would be glad to improve it if more details are desirable or different structure is advised.

More details are needed on bioinformatic and statistical analysis approaches used for the scRNA-seq data.

We re-wrote the method section related to bioinformatics and statistical analysis providing more details.

Information on sample size and power calculations is lacking and should be included.

Single cell data presented in this manuscript is derived from two MT samples, one from a patient with schizophrenia and one from an individual without psychiatric disorders. Similarly, single cell data from cell lines also derived from two individuals, one from patient with schizophrenia and one from individual without psychiatric disorders. This is clearly stated in the first paragraph of the Results section. Thus, every group is presented by one sample, not allowing to assess the distribution of gene expression within the groups and and other critical parameters such as biological variance.

Describe any measures taken to reduce technical biases in sample processing and analysis.

We made efforts to process two samples the same way. However, analysis of technical biases and confounding factors make sense when more than two samples are used, and when statistical analysis of data is required.

Results:

The order of presenting results could be improved for clarity - consider grouping by sample type.

Use figures/tables to summarize key results instead of just describing in text.

We have altered legends to tables and figures, made them more explanatory.

Report exact p-values and confidence intervals for group comparisons.

We have not performed statistical comparison of data between groups with exception of comparison of gene expression between clusters with the goal of identification of marker genes.

Discussion:

Relate current findings to previous studies identifying transcriptional profiles of nasal epithelium or embryonic brain cells.

Large part of this study and most of Discussion section is devoted to relationship of CNON transcription profiles to transcription profiles of cells from olfactory neuroepithelium and cells from cortical region of embryonic brain. It is not clear which points the reviewer would like us to discuss in addition to those already present in the manuscript.

line 356, snp polymorphisms associated with different phenotipic xpression should be discussed . discuss and cite doi:10.1136/bjophthalmol-2021-319756. and doi:10.1111/coa.13870.

It must be a mistake to advise citing doi:10.1136/bjophthalmol-2021-319756; the article discuss association of SIX1-SIX6 polymorphisms with peripapillary retinal nerve fibre layer thickness in children, and our study is not related to this topic. The second recommendation, doi:10.1111/coa.13870, is a review of single-nucleotide polymorphism in chronic rhinosinusitis, while our study is related to schizophrenia or, more broadly, to brain disorders.

Elaborate on limitations related to sample heterogeneity, sample size, technical factors etc.

Given that we are presenting data from one sample per group, discussion sample heterogeneity and sample size seems inappropriate.

Suggest future research directions to build on these findings.

We added future direction section to Discussion.

Conclusions:

Restate purpose and key findings briefly to lead into conclusions.

Emphasize implications of the findings for modeling neurodevelopmental processes.

Suggest applications of the CNON model based on the results.

I think most of the suggestions were already reflected in the original text. We modified conclusion section slightly to further emphasize potential use of CNON.

Reviewer 2 Report

In their research manuscript entitled "Cultured Mesenchymal Cells from Nasal Turbinate as a Cellular Model of the Neurodevelopmental Component of  Schizophrenia Etiology", Victoria Sook Keng Tung and colleagues have performed single cell-RNA sequencing in cells derived from olfactory neuroepithelium of schizophrenia patients and controls (CNON cells). They compare their scRNAseq CNON data to scRNAseq data from biopsy samples and published scRNAseq data of embryonic brain. The manuscript addresses an important aspect in schizophrenia research, i.e. the identification and characterization of valid cellular types that can be accessed easily and followed through the progression of the disease in patients. The authors provide compelling evidence that CNON cells originate from a single cell type of olfactory neuroepithelium that closely matches mesenchymal stem cells.

I have several questions/omments to address to the authors, aiming at the improvement of the discussion and the presentation of the results in the manuscript.

1.  An aspect that is not clearly discussed in the text relates to other cells derived form the olfactory neuroepithelium that have been used in schizophrenia research. For example, researchers have utilized olfactory neuroepithelium neurosphere-derived cells to hihlight differences between schizophrenia patients and controls (e.g. https://www.ncbi.nlm.nih.gov/pmc/articles/PMC4930119/). What would be the relationship of those neurosphere-derived cells, if any, with the CNON cells that the authors study? Perhaps authors should comment on that, at least briefly.

2. Introduction, lines 37-38: the authors should elaborate more on the use of patient-derived cell types. They should point out other peripheral cells like PBMCs, as well as other olfactory neuroepithelial cells (see comment above). Also this section requires accurate referencing.

3. Results, line 166: The authors should provide some details about the patient and control's samples. Ideally, age, sex, medication status, etc., should be disclosed.

4. Figure 1 and FIgure 2 graphs: the lettering in axes in too small to be seen even at high zoom. Please enlarge

5. Results, lines 324-325: the authors suggest a higher apoptosis rate to account for the slow growth of CNON-SCZ cells. They would validate their claim if this could be tested with a specific cellular assay. Would that be doable?

6. DIscussion: It seemed to me a bit unorthodox that the authors continue to introduce new results (Tables 4-5 and Figure 3) in the DIscussion section.  Status of Wnt and Notch signaling in CNON cells (Table 5) and MC status of CNON cells (Table 4), or morphology of CNON cells (FIgure 3), are important feautures but they should have been introduced earlier in the Results. Perhaps some of these results could be presented as Supplementary. 

7. FIgure 3: The authors should provide better resolution images to support their claims about changes of CNON cells in different culture conditions. Ideally, images should be presented with the same scale bar to make the comparison evident. The scale bar is present in A and C but it is incomprehensible. What is the scale bar for A-C?

8. DIscussion, lines 434-437: Sentence is complicated. Please rephrase

-

Author Response

  1. An aspect that is not clearly discussed in the text relates to other cells derived form the olfactory neuroepithelium that have been used in schizophrenia research. For example, researchers have utilized olfactory neuroepithelium neurosphere-derived cells to hihlight differences between schizophrenia patients and controls (e.g. https://www.ncbi.nlm.nih.gov/pmc/articles/PMC4930119/). What would be the relationship of those neurosphere-derived cells, if any, with the CNON cells that the authors study? Perhaps authors should comment on that, at least briefly.

CNON is a specific cell type, which expression profile is stable during at least several passages. Other protocols may result in a mixture of different cell types (Lampinen et al., cited in the manuscript), or they may change expression profile over time, the latter is a special concern when neurosphere-derived cells are used. There is data (Tome et al, cited in the manuscripts) indicating that two different types of neurospheres can be produced from ON biopsies, one is likely originated from basal cells, and another one is likely originated from mesenchymal cells. In series of studies by French group (Delorme et al. cited in the manuscript) there is evidence that neurosphere-derived cells from ON lamina propria biopsy samples are similar to CNON and exhibit mesenchymal properties. However, olfactory mucosa likely produces different neurosphere types. We modified corresponding text in the discussion section to better address the complexity of different cellular models derived from human ON.

  1. Introduction, lines 37-38: the authors should elaborate more on the use of patient-derived cell types. They should point out other peripheral cells like PBMCs, as well as other olfactory neuroepithelial cells (see comment above). Also this section requires accurate referencing.

Our study is focused on cellular models to study neurodevelopmental aspects of brain disorders, which we stated in the first paragraph of Introduction. There is no reason to believe that blood-based cellular models could be helpful in this regard, while widening the discussion to all patient-derived cellular models will de-focus the study. Relevance of blood-based cellular models to brain disorders often raises questions. Some time ago I performed comparative analysis of differentially expressed genes found in several large studies of schizophrenia and found that overlap between findings is even less than expected by chance. I believe many researchers are skeptical about results of studies based on blood-based cellular model, and we would not like to bring that model into comparison with our approach.

We added references to some notable studies using IPSCs or IPSC-derived neuronal cells to study schizophrenia.

  1. Results, line 166: The authors should provide some details about the patient and control's samples. Ideally, age, sex, medication status, etc., should be disclosed.

We provided information about patients and individuals from the control group used in the study. We do not have information about medications used by patients.

  1. Figure 1 and FIgure 2 graphs: the lettering in axes in too small to be seen even at high zoom. Please enlarge

We re-made the figures. It seems that part of the problem is that resolution of all figures was reduced during preparation of the draft as it is lower than the original figures we provided.

  1. Results, lines 324-325: the authors suggest a higher apoptosis rate to account for the slow growth of CNON-SCZ cells. They would validate their claim if this could be tested with a specific cellular assay. Would that be doable?

Yes, it is doable, and it is in our plans. However, simple assay would not provide sufficient information to test the hypothesis. Apoptosis rate is likely dependent on cell density and correlated with cell proliferation rate. We plan to assess these two parameters during cell growth, performing apoptotic assay and counting cells at the same time in monolayer at different cell densities. We also would like to be able to perform single cell transcriptomic analysis at different cell densities to identify genes which possibly trigger apoptosis. Finally, we would like to test the effects of some potentially critical genes, such as BIRC5, knocking down their expression (which we have done in one experiment already) to validate the hypothesis. Together, this data will provide a more comprehensive and more convincing picture of apoptosis - cell proliferation relationship and possible involvement of this axis in the etiology of schizophrenia.

  1. DIscussion: It seemed to me a bit unorthodox that the authors continue to introduce new results (Tables 4-5 and Figure 3) in the DIscussion section. Status of Wnt and Notch signaling in CNON cells (Table 5) and MC status of CNON cells (Table 4), or morphology of CNON cells (FIgure 3), are important feautures but they should have been introduced earlier in the Results. Perhaps some of these results could be presented as Supplementary.

We restructured the manuscript and moved data with the results from Discussion section into Results section.

  1. FIgure 3: The authors should provide better resolution images to support their claims about changes of CNON cells in different culture conditions. Ideally, images should be presented with the same scale bar to make the comparison evident. The scale bar is present in A and C but it is incomprehensible. What is the scale bar for A-C?

Original images were provided with higher resolution and legible scale bar label. Probably a draft version for review has simplified image processing. We provided a different set of images taken at the same magnification.

  1. DIscussion, lines 434-437: Sentence is complicated. Please rephrase

We re-phrased the sentence.

Round 2

Reviewer 2 Report

The authors have responded to my previous comments.